# Robust Image Classification with Cognitive-Driven Color Priors

**Peng Gu** [1,2] iD **, Chengfei Zhu** [2] **, Xiaosong Lan** [2,3] **, Jie Wang** [1,2] **and Shuxiao Li** [1,2,]*

[1]   School of Artificial Intelligence, University of Chinese Academy of Sciences, Beijing 100049, China; gupeng2018@ia.ac.cn (P.G.); wangjie2019@ia.ac.cn (J.W.)

[2]   Institute of Automation, Chinese Academy of Sciences, Beijing 100190, China; chengfei.zhu@ia.ac.cn (C.Z.); lanxiaosong2012@ia.ac.cn (X.L.)

[3]   Innovation Academy for Light-Duty Gas Turbine, Chinese Academy of Sciences, Beijing 100080, China

[*]   Correspondence: shuxiao.li@ia.ac.cn

**Abstract:** Existing image classification methods based on convolutional neural networks usually use a large number of samples to learn classification features hierarchically, causing the problems of over-fitting and error propagation layer by layer. Thus, they are vulnerable to adversarial samples generated by adding imperceptible disturbances to input samples. To address the above issue, we propose a cognitive-driven color prior model to memorize the color attributes of target samples inspired by the characteristics of human memory. At inference stage, color priors are indexed from the memory and fused with features of convolutional neural networks to achieve robust image classification. The proposed color prior model is cognitive-driven and has no training parameters, thus it has strong generalization and can effectively defend against adversarial samples. In addition, our method directly combines the features of the prior model with the classification probability of the convolutional neural network, without changing the network structure and its parameters of the existing algorithm. It can be combined with other adversarial attack defense methods, such as various preprocessing modules such as PixelDefense or adversarial training methods, to improve the robustness of image classification. Experiments on several benchmark datasets show that the proposed method improves the anti-interference ability of image classification algorithms.

**Keywords:** adversarial samples; color prior model; image classifification

## 1. Introduction

In recent years, deep neural networks (DNN) have achieved outstanding performance in artificial intelligence fields such as image processing [1], natural language processing [2], and speech recognition [3]. However, some studies have shown that DNN has inherent blind spots, making it susceptible to be attacked by blurred samples [4–6]. In 2013, Szegedy et al. [7] first discovered an anti-intuitive phenomenon in the field of image classification: by adding human undetectable perturbations to test samples, deep learning based classifiers can probably misclassify with high confidence when tested. They call such input samples with slight perturbations as adversarial samples. Compared with noisy samples, adversarial samples are deliberately designed and are not easy to detect, which can lead to higher false prediction confidence of the classifier. More seriously, opponents can design adversarial samples in application scenarios, which may cause unnecessary confusion [8] in areas such as identity recognition or autonomous driving.

So far, two main methods have been proposed to defend against adversarial samples. The first type is to enhance the robustness of the classifier itself. Adversarial training [9] and defensive distillation [10] belong to this category method. Adversarial training adds adversarial samples to the training data to

retrain the classifier to achieve the purpose of defensing adversarial samples. The defensive distillation method uses distillation to train two deep neural DNN models connected in series to improve the robustness of the classifier. These methods require a large number of adversarial samples and normal samples, which consumes training time and hardware resources, and the trained defense model usually does not have strong generalization ability. The other type is a variety of pre-processing modules, such as the PixelDefense method proposed by Song et al. [11], which converts the adversarial samples into clean images and inputs the converted images to the classifier for prediction, this category of method only needs to pre-process the test samples and does not change the overall structure of the model, but the false positive rate and false negative rate on the prediction results are large.

The visual attention mechanism originates from the study of human vision and has been extensively explored in various scenarios [12,13]. In human cognitive science, the information processing capabilities of different parts of the retina are different. In order to make reasonable use of limited visual information processing resources, various objects in the visual scene compete with each other in order to selectively focus on specific parts of the visual area. There are two different methods of visual attention: one is stimulus-driven attention, which attracts locally distinctive parts of the image for people's attention; The other is task-driven attention, which achieves top-down modulation by increasing the significance of expected locations or features of the target of interest [12].

Inspired by the task-driven visual attention mechanism, this paper proposes a cognitive-driven color prior model for robust image classification. The overall flowchart of the proposed algorithm is shown in Figure 1. It mainly includes:

(1) "Off-line memory" stage: The aim is to establish color prior models of typical category images based on visual attention mechanism and store them. For the convenience of memory and mapping, patterns are used to represent local features of the image, which is essentially a finite one-dimensional discrete number. We use a tuple of {pattern, saliency} to represent the prior model. For a given training image set, the saliency of high frequency patterns is enhanced by accumulating the pattern occurrences of samples within the same category, and the saliency is further enhanced or reduced by competing the relative frequencies between different categories. The tuple of {pattern, saliency} for each category represents the degree of prior correlation between the target pattern and the category, which needs to be calculated once and saved in the table. In order to improve the calculation efficiency in the mapping stage, we only save the color prior model of some selected categories with significant color characteristics.

(2) "On-line mapping" stage: Inspired by human conditioned reflex characteristics [12], we obtain the color saliency features of the current image in the form of a lookup table, and fuse them with neural network features to prevent interference from adversarial samples. Based on the established cognitive-driven color prior model, the saliency feature of the test image is obtained by the retrieval method [14], and is regarded as a feature map. Then, block average pooling and part-whole mapping are performed to obtain color-prior-based part features and classification probabilities, respectively. Finally, classification probabilities from color priors and DNN are merged, as if adding a neural network with human-like memory characteristics.

Experiments prove that the proposed method improves the anti-interference ability of neural networks. Its main advantages and novelties are as follows:

- We propose to use a tuple of {pattern, saliency} to represent the prior model, so that the saliency feature of the test image can be quickly obtained by looking up the table.
- The proposed color prior model comprehensively considers pattern frequencies within the same category and the competition between different categories without training parameters, thus has certain anti-interference ability.
- We fuse color prior model and DNN at the classification probability level, which does not affect the overall structure of the neural network, and can be used in combination with the respective preprocessing modules or adversarial training methods.

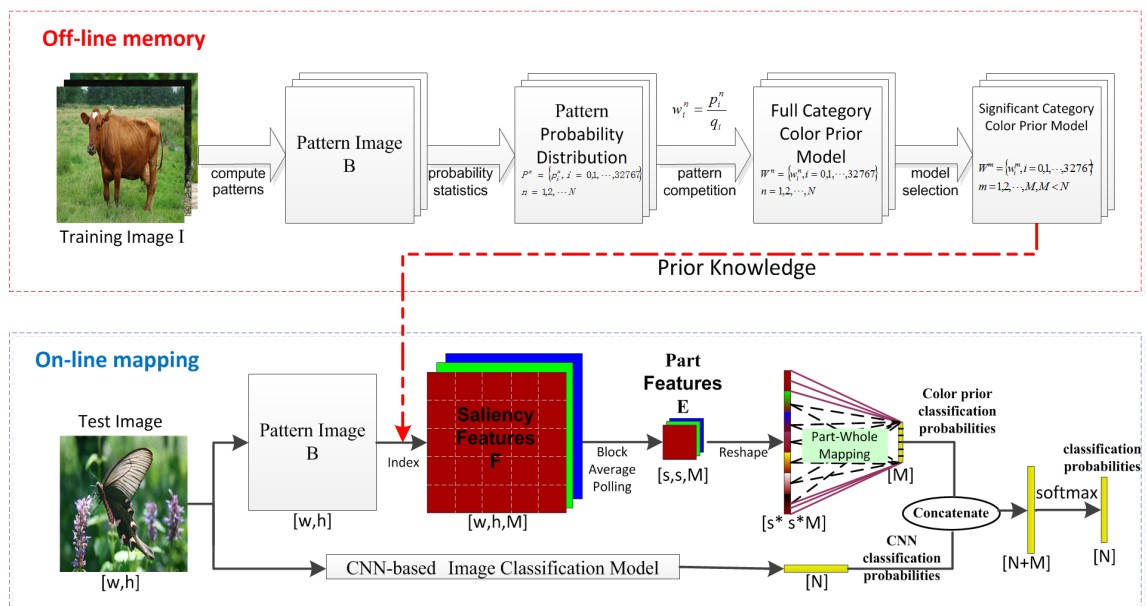

**Figure 1.** Overall architecture of image classification method combined with color prior model.

The rest of this paper is organized as follows—Section 2 introduces related work, Section 3 and Section 4 detail "off-line memory" stage and "on-line mapping" stage of the proposed method, respectively. Experiments to verify the defensive effects of the color prior model and its ablation studies are illustrated in Section 5. Finally, Section 6 gives the main conculsions.

## 2. Related Work

This section mainly introduces related work from three aspects: image classification networks, adversarial samples attack methods, and adversarial samples defense methods.

### 2.1. Image Classification Network

Since AlexNet [15,16] won the ILSVRC competition, convolutional neural networks have become the mainstream method for image classification, more and more classical networks continue to spring up [17–19]. GoogLeNet [20] propose the inception structure and increases the diversity of features. VGG [21] explores the relationship between network depth and classification accuracy. ResNet [1] solves the problem of gradient explosion and gradient disappearance caused by the deepening of the network structure. DenseNet [22] effectively reduces the scale of network parameters and suppresses over-fitting. In terms of simplifying neural networks, Mobilenet [23] is a lightweight neural network proposed by Google for mobile devices, which effectively reduces the amount of parameters and calculations. ShuffleNet [24], PeleeNet [25] and ThunderNet [26] enable the network model to be further optimized and become smaller and faster.

### 2.2. Adversarial Samples Attack Methods

Adversarial samples can be divided into three categories by the attack methods—(1) Gradient-based attack methods [5,27,28]. These approach require all the information of the model, including network structure, model weight parameters, and so forth. They calculate the derivative of the model to the input and determine the gradient direction, and then superimpose it on the input image to generate adversarial samples. FGSM proposed by Kurakin in 2016 is a typical gradient-based attack method [5]. (2) Methods based on iterative optimization. These methods are usually optimized in the direction of interference intensity and recognition confidence [29–34]. The effective interference noise is found by iterative calculation, and the interference intensity generally increases with the increase of iteration times. Typical methods include the L-BFGS method proposed by Szegedy et al. [29], the JSMA attack

method proposed by Papernot, and so on. Methods based on iterative optimization are also shifting from white box attacks to black box attacks [33]. In addition to the white box attack methods mentioned above, there is also the One Pixel Attack black box attack method proposed by Su [31]. (3) Mixed attack methods. In fact, many attack methods simultaneously use gradient information and iterative optimization to generate adversarial examples [34–36].

### 2.3. Adversarial Samples Defense Methods

Adversarial samples defense methods can be divided into three categories: (1) Modify training data. Such as data compression, confrontation training and other methods [9,37–39]. These methods generally add disturbance data to the training data to train the network and do not change the network model. Modify training data is a simple and effective means for model optimization, the disadvantages are high computational cost, passive response and so on. (2) Modify the network model. Such as gradient shielding, modifying the loss function, modifying the activation function, and so forth [40–44]. Defensive distillation method proposed by Papernot [44] is a typical method of gradient shielding, which can make the output of the model more uniform to resist small interference. (3) Additional network structure. This type of methods mainly adds new network layer to the original network [45–47]. For example, Akhtar [46] proposed to add a pre-input layer to the model, so that the disturbance of the adversarial sample could be disappeared in the pre-training layer as much as possible.

## 3. "Off-Line Memory" Stage

This stage aims to establish color prior models for typical categories. Next, we will introduce each part of the algorithm in detail, including model representation, model estimation and model selection.

### 3.1. Prior Model Representation

We use patterns to represent local features of images, so that memory and mapping can be achieved through probabilistic analysis and table lookup, respectively. Let $I \in R^{w \times h \times 3}$ be the input image, where $w$ and $h$ are the image width and image height, respectively. We use the pattern operation $\Phi$ to transform the input image to a pattern image $B \in Z^{w \times h}$ (discrete integer space), which can be expressed as:

$$B = \Phi(I) \tag{1}$$

The pattern operation $\Phi$ can be the serialization of color features or a texture feature operator such as local binary pattern. This article mainly focuses on the serialization of RGB color features. Specifically, we quantize and concatenate R,G,B values to obtain the color pattern at pixel-level. Quantizing can reduce the total number of patterns and we quantizes each channel to [0,32) to achieve a good balance between discrimination and anti-interference ability. As a result, the encoded color pattern occupies 15 bits per pixel, which is in the range of [0,32767]. Provided $I_R, I_G, I_B$ are the R,G,B values of a pixel, its color pattern can be calculated by:

$$\begin{aligned} B_c = \Phi_c(I) &= Con\{I_R \gg 3, I_G \gg 3, I_B \gg 3\} \\ &= (I_R \gg 3) \ll 10 + (I_G \gg 3) \ll 5 + (I_B \gg 3), \end{aligned} \tag{2}$$

where $\gg$ and $\ll$ are shift operators.

Converting RGB images to pattern images is beneficial for memory and mapping. Memory can be simply done by summing up pattern occurrence frequencies in training images, from which color prior saliency can be learned and stored. When recognizing the world, the memorized priors can be picked up by indexing, which is similar to human conditioned reflex. Therefore, the color prior model of the $n$-th class can be expressed as:

$$W^n = \{w_i^n, i = 0, 1, \cdots, 32767\}, n \in [1, N], \tag{3}$$

where $N$ is the total number of interested classes, and $w_i^n$ is the prior saliency value for the $i$-th color pattern of the $n$-th class.

*3.2. Prior Model Estimation*

In the scenes observed by humans, target objects within the same class tend to be aggregated together due to their similarities in color, intensity, and texture. Let $I^{n,k}$ be the $k$-th training image of the $n$-th class, we can obtain its pattern probability distribution(PPD) by calculating the pattern histogram and then normalizing it:

$$\begin{aligned}P^{n,k} &= Normalize\{Hist\lfloor\Phi_c(I^{n,k})\rfloor\}\\ &= \{p_i^{n,k}, i = 0, 1, \cdots, 32767\}.\end{aligned} \tag{4}$$

The PPD of each class can be obtained by continuously "remembering" the PPD of all training images within the same class, which can be approximately estimated by the mean value:

$$P^n = \left\{\frac{1}{N_k} \times \sum_{k=1}^{N_k} p_i^{n,k}, i = 0, 1, \cdots, 32767\right\}. \tag{5}$$

For a system that can learn online, we first obtain the PPD $M_T^n$ of the new sample set at moment $T$ according to Formula (5), and then linearly update the existing model:

$$P_T^n = \left\{(1 - \eta) \times p_{i,T-1}^n + \eta \times m_{i,T}^n, i = 0, 1, \cdots, 32767\right\}, \tag{6}$$

where $\eta$ is the forgetting factor. After using Formula (5) or Formula (6) to obtain the PPD of all categories, we can see significant differences between different categories. However, such a model has not yet introduced a visual competition mechanism, leaving room for improvement in saliency expression.

There are always lots of different types of objects in the visual scene. At this time, the focus of human observation is usually driven by attribute competition among them. If pattern $i$ of class $n$ occurs more frequently than that of the natural scene or other classes, it means that this pattern is "competitive" and should be enhanced, otherwise it should be suppressed. We use the mean of $P^n$ of all category targets to approximate the mixed PPD $Q$ of natural scenes and other classes:

$$Q = \left\{q_i = \frac{1}{N} \times \sum_{n=1}^{N} p_i^n, i = 0, 1, \cdots, 32767\right\}. \tag{7}$$

Then the color saliency model of class $n$ can be expressed as:

$$W^n = \left\{w_i^n = \frac{p_i^n}{q_i}, i = 0, 1, \cdots, 32767\right\}, \tag{8}$$

where $w_i^n$ reflects the degree of prior correlation between the $i$th color pattern and the $n$-th object class, which can be directly used for saliency estimation.

*3.3. Prior Model Selection*

Based on a given training image set, the full category color prior model $\{W^n, n = 1, 2, \cdots, N\}$ can be built according to the aforementioned method. However, not every object class has significant color characteristics. In order to reduce the interference of unreliable color features and improve the computational efficiency of the inference phase, we only retain the color prior model of typical classes with significant color characteristics:

$$\{W^m, m = 1, 2, \cdots, M\}, M \leq N. \tag{9}$$

Based on the test set of images and $W^n$, we calculate the prior success rate (PSR) for each category to select $W^m$. Firstly, we use Equation (2) to obtain the pattern image $B \in Z^{w \times h}$ for each test image, and generate the color prior saliency feature image $\{F^n, n = 1, 2, \cdots, N\}$ by indexing the pattern number's weight in $W^n$. Then, the prior classification probability $\{f^n, n = 1, 2, \cdots, N\}$ is obtained by using global average pooling on $F^n$. Finally, the category corresponding to the maximum value of $f^n$ is selected as the prediction label of the test image. If the prediction label is consistent with the ground-truth label, it is determined that the test image is successfully classified according to the prior, otherwise we declare that it is failed. After evaluating on all test images, the PSR of all categories $\{s^n, n = 1, 2, \cdots, N\}$ can be obtained. We sort $s^n$ in descending order, only keeping the top $M$ categories for online calculation.

## 4. "On-Line Mapping" Stage

This stage is mainly used to calculate the class probability of the test image. We first obtain saliency feature through color prior model mapping. Then, block average pooling and part-whole mapping are designed to generate color prior classification probabilities, which are finally combined with those of convolutional neural networks (CNN) to achieve robust image classification.

### 4.1. Prior Saliency Feature Generation

For the input image $I \in R^{w \times h \times 3}$, the pattern operation $\Phi$ is used to map it to the pattern image $B \in Z^{w \times h}$. For each pixel $x \in R^2$ in the pattern image, $B(x) \in Z$ is the color pattern of the pixel. Taking $B(x)$ as the index and reading the corresponding prior saliency values $\{w_{B(x)}^m, m = 1, 2, \cdots, M\}$ from the color prior model $W^m$ as its feature, we get $M$ prior features for each pixel of the input image. Finally, the prior saliency of all pixels are organized in the form of a multi-channel image to obtain a priori saliency feature $F \in R^{w \times h \times M}$ with $M$ channels. We treat it as a feature map and perform subsequent operations.

### 4.2. Block Average Pooling

The prior saliency feature extracts local information, which is discriminative but not robust. On the other hand, directly processing the prior saliency feature by the global average pooling to get the global feature is robust but not discriminative. In fact, the target is usually composed of typical parts. Some parts reflect the typical features of the category (strong saliency), while the others are susceptible to background interference and have poor intra-class consistency (weak saliency). Thus, making the best of part features can achieve better target recognition performance.

We propose a block average pooling technique to extract target part features. Firstly, each prior saliency feature $F^m$ is divided into $s \times s$ feature blocks. Then, global average pooling operation is applied to each block instead of the whole image to get totally $s \times s$ part features of $F^m$. Therefore, through the block average pooling technology, we can obtain the part features $E \in R^{s \times s \times M}$ based on the saliency feature $F \in R^{w \times h \times M}$. Finally, we reshape it to a one-dimensional vector of length $s \times s \times M$.

### 4.3. Part-Whole Mapping

From the generation process of prior saliency feature image and prior part features, it can be seen that there is a strong correspondence between the prior part features $E \in R^{s \times s \times M}$ and the object class, so the "part-whole mapping" is mainly based on intra-class mapping. In addition, there are some "weak associations" between features of different classes, which can provide a small amount of additional information for each other.

We use a fully-connected layer of shape $[s \times s \times M, M]$ to implement the part-whole mapping, and obtain M-dimensional color prior classification probabilities. In order to reflect the above-mentioned "part-whole mapping" characteristics, we add an additional positive constant to the mapping connections within the same class on the basis of random initialization to enhance its importance.

*4.4. Fusion with CNN Features*

CNNs are prone to the problems of over-fitting and error propagation layer by layer, so they are very susceptible to interference from adversarial samples. This article uses human memory characteristics to obtain the color saliency to improve the classification robustness. Given a test image, we can acquire $M$ color priori classification probability features according to the above procedures. On the other hand, $N$ classification probabilities can be extracted from the test image through a CNN, where $N$ is the total number of interested classes and $M \leq N$. Finally, these two complementary features are fused, which enhances the anti-interference ability of the CNN as well as improves the classification accuracy.

## 5. Experiments

*5.1. Datasets*

Three common datasets are used to verify the performance of the proposed algorithm, with TOP1 recognition rate as the measure. We use ImageNet [27] to do ablation analysis. In order to accelerate the validation process, we only select 20 commonly used object classes from it. For comparative studies, we use another two datasets of CIFAR-10 [48] and Fashion-mnist [49] to verify the effectiveness of the proposed method.

We use fast gradient sign method (FGSM) [50] to generate adversarial samples. Although the adversarial sample only adds some subtle disturbances, it is very deceptive to the neural network, resulting in a greatly reduced recognition ability of the neural network, as shown in Figure 2. The adversarial samples are generated by:

$$X' = X + \varepsilon \bullet sign(\bigtriangledown x J(X,Y)), \tag{10}$$

where $X'$ is the generated adversarial sample, $X$ is the original image, $Y$ is the ground-truth label, and $J(X,Y)$ is the loss function. The interference factor $\varepsilon$ is selected to control the disturbance magnitude, and we select six values to generate adversarial samples with different degrees of interference, including 0.5, 1, 1.5, 2, 2.5, 3.

- ImageNet: This dataset is one of the most famous datasets in the image processing world. Each category contains 1300 training images and 50 test images. The generated adversarial examples are shown in Figure 3.
- CIFAR-10: This dataset contains 60,000 color images with a size of $32 \times 32$ and a total of 10 categories. There are 5000 training images and 1000 test images per category. The generated adversarial examples are shown in Figure 4.
- Fashion-mnist: This dataset is an improved version of Mnist, which covers frontal fashion pictures of a total of 70,000 different products from 10 categories. It contains 60,000 training images and 10,000 test images, each of which is a $28 \times 28$ gray-scale image. The generated adversarial examples are shown in Figure 5.

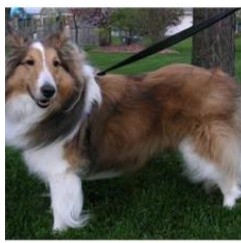　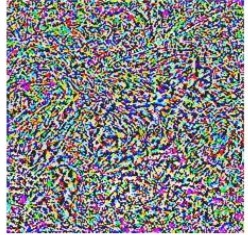　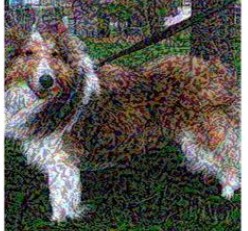

Dog：96.05%　　　　　　　　　　　　　　　　　　　　　　　　Cock：100.00%
(a) Original Image　　　　　　(b) Disturbance　　　　　　(c) Adversarial Examples

**Figure 2.** Examples of adversarial examples.

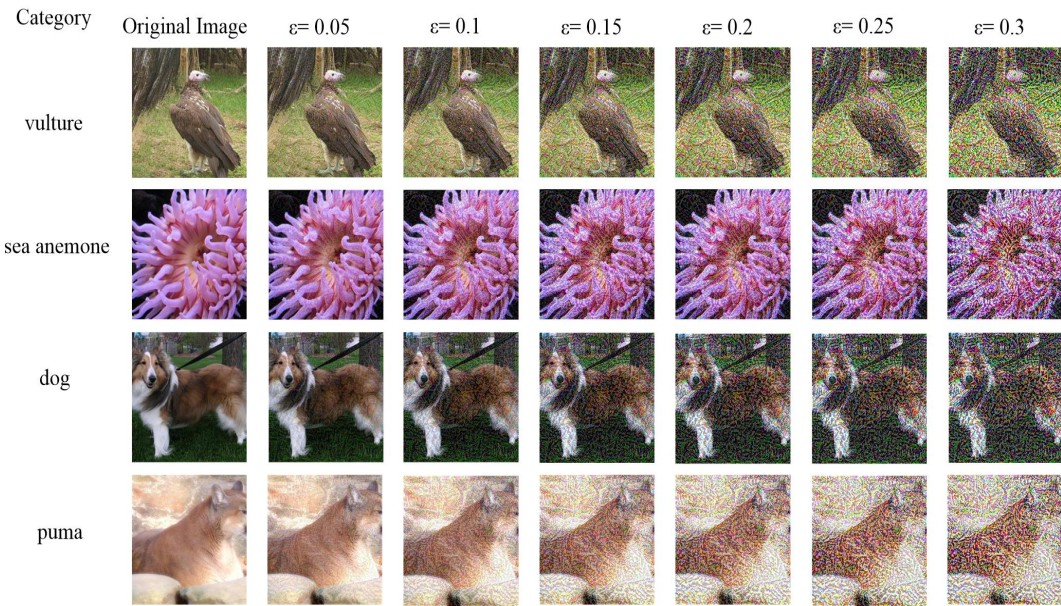

**Figure 3.** Generated adversarial examples from ImageNet dataset.

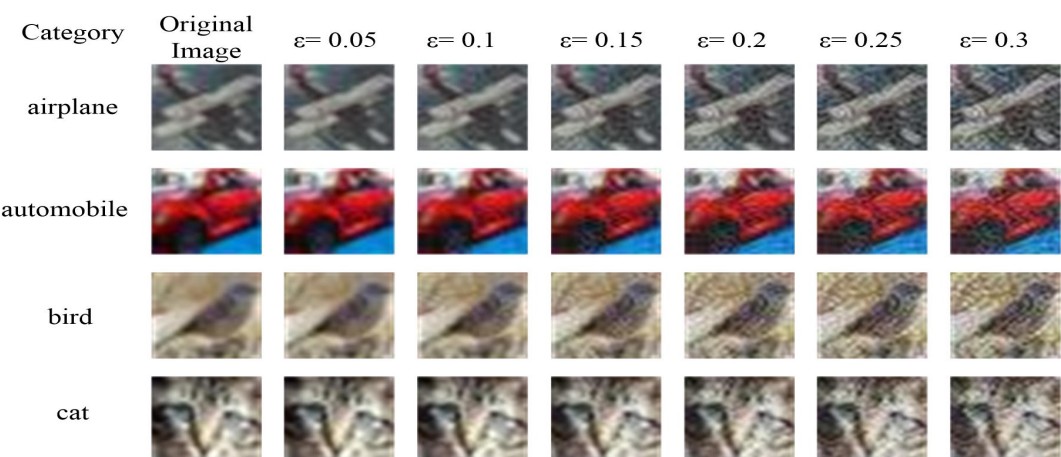

**Figure 4.** Generated adversarial examples from CIFAR-10 dataset.

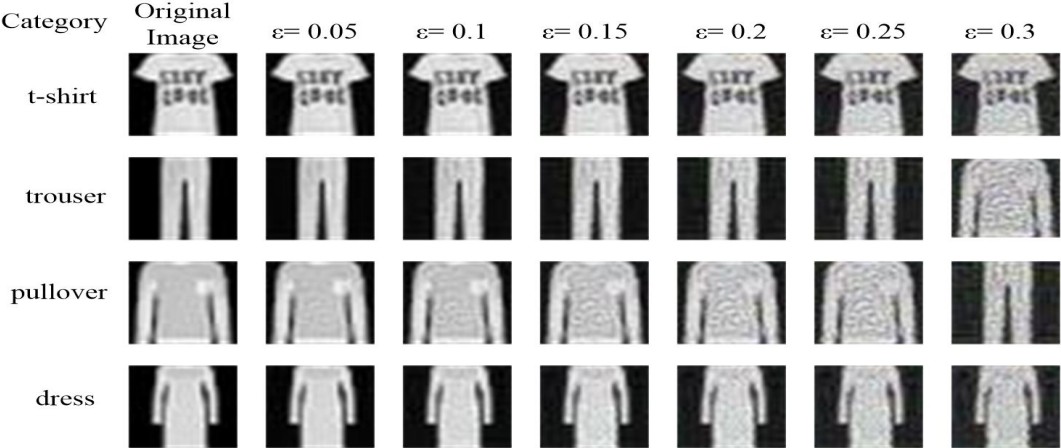

**Figure 5.** Generated adversarial examples from Fashion-mnist dataset.

## 5.2. Implementation Details

Color priori model involves two important parameters: the number of part feature blocks $s \times s$ and the number of significant categories $M$. For the test image, the color prior saliency maps of N

categories are generated by indexing the color saliency model, which are shown in Figure 6. In order to reduce the interference of unreliable color features and the burden of neural network, we use PSR mentioned in Section 3 to establish the saliency category color prior model, which is composed of the most distinguishing $M$ classes selected from the $N$ classes full category color prior model. Through experimental analysis on the ImageNet, we take $s = 5, M = 14$. Thus, for each test image, a total of 350 ($5 \times 5 \times 14$) prior part features belong to 14 categories are extracted. For CIFAR-10 and Fashion-mnist, we take $M = 6, 8$ respectively and $s$ is still 5.

For part-whole mapping, the parameters are firstly initialized randomly. Then, the mapping connections within the same class are increased by 0.8. The network weight parameters in the fusion phase are randomly initialized. We chose PeleeNet [25] to do ablation experiments. In the comparison experiments, we choose another six classic classification networks, including ResNet50 [1], VGG16 [21], MobileNetV2 [51], DenseNet161 [22], InceptionV3 [52], SqueezeNet [53]. We use a training cycle of 240 and a batch of 32. Other hyper-parameters are consistent with open source projects [25]. After training, we solidify all the parameters of the CNN and directly use the classification probability features of its output.

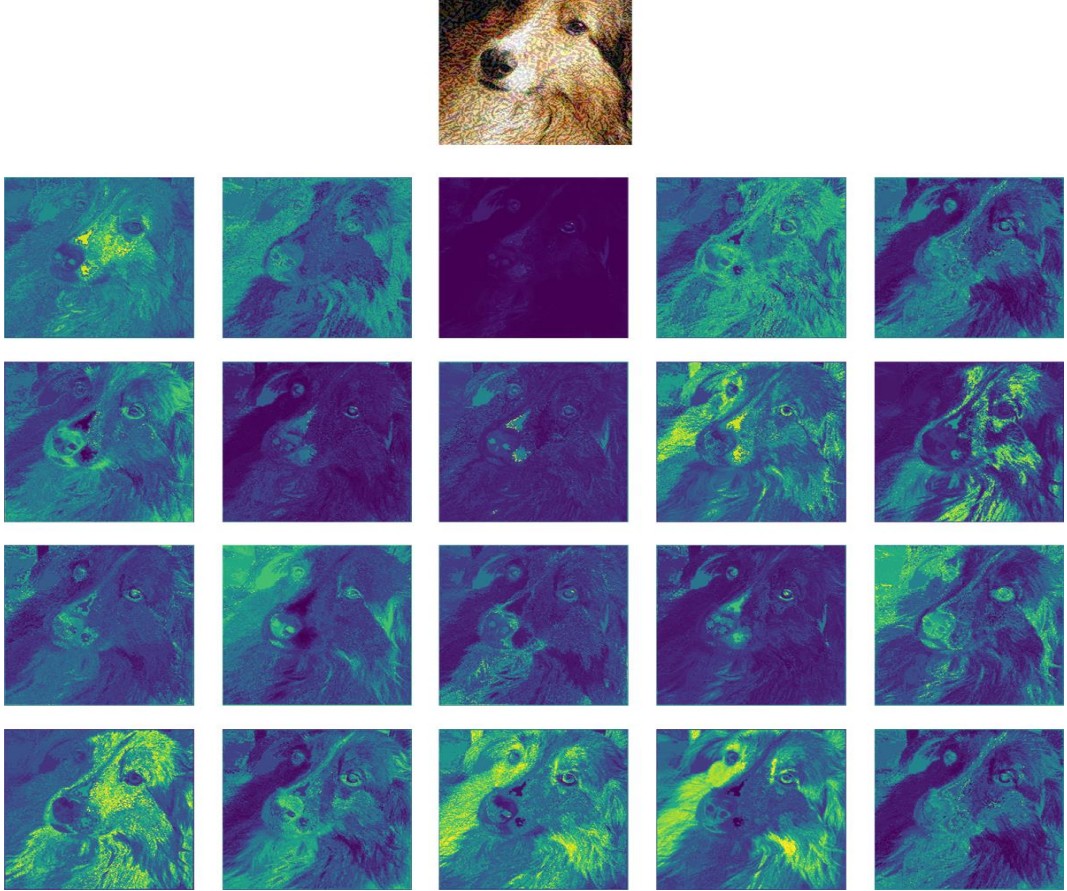

**Figure 6.** Test image and saliency features.

*5.3. Ablation Study*

We conduct experiments according to whether the following designed ideas are included:

- Whether to include a part model: The alternative is to use global average pooling instead of block average pooling.
- Whether to perform prior model selection: The alternative is to use the full-class color prior model instead of the salient class color prior model for online calculation.

- Alternatives to model selection: An alternative way beyond PSR is using the Information Entropy (IE) for model selection. If the PPD contains several significant peaks, it means that the color characteristics of the object class is prominent. Otherwise, if the distribution is disorderly, it means that the color characteristics of the object class is not obvious. This characteristic can be measured by information entropy. We calculate IE of the PPD for each class and chose M object classes with largest IE values for online calculation.

In order to verify the effectiveness of the color prior model, we tested PeleeNet [25] . The results are shown in Table 1, the interference factor $\varepsilon$ controls the interference intensity. Compared with the original framework PeleeNet [25], it can be found that the color prior model improves the recognition rate of CNN. The improvement is more obvious in the case of adversarial attack. It shows that cognitive-driven experience features can enhance the anti-interference ability and performance of neural networks. The prior saliency feature extracted by global average pooling is not discriminative, we proposed the PART model using Block Average Pooling technology to obtain robust and distinguishing target part features. In Table 2, it is shown that the part model improved the recognition rate under interference, which could be increased by 1.7 percents at most. In order to verify the importance of model selection and the difference between different selection methods, we conducted comparative experiments, the results are shown Table 3. It can be seen that the PSR-based model selection scheme is better than the IE-based model selection scheme, indicating that the model selection method is an important factor affecting the performance of the algorithm and deserves further study. The PSR-based model selection strategy improves the overall recognition rate while reducing the amount of calculation by about 33%. This allows the color prior features and CNN features to use part of the CPU resources and GPU for simultaneous calculations, ensuring the efficiency of the algorithm.

The results of ablation experiments are comprehensively shown in Table 4, the optimal value and sub-optimal value of the experimental effect are marked with bold and underlined respectively.

**Table 1.** Comparison of the original frame with or without the color prior model (datasets:imagenet).

| | TOP1 Recognition Rate under Different FGSM Interference Amplitudes (%) | | | | | | |
| --- | --- | --- | --- | --- | --- | --- | --- |
| | $\varepsilon = 0$ | $\varepsilon = 0.5$ | $\varepsilon = 1$ | $\varepsilon = 1.5$ | $\varepsilon = 2$ | $\varepsilon = 2.5$ | $\varepsilon = 3$ |
| PeleeNet | 93.0 | 18.4 | 12.3 | 10.1 | 9.7 | 8.8 | 8.2 |
| PeleeNet+ColorPriors | 93.1 | 20.0 | 12.5 | 11.0 | 9.9 | 9.5 | 9.2 |

**Table 2.** Comparison of color prior models with or without part model (datasets:imagenet).

| | Part Model | TOP1 Recognition Rate under Different FGSM Interference Amplitudes (%) | | | | | | |
| --- | --- | --- | --- | --- | --- | --- | --- | --- |
| | | $\varepsilon = 0$ | $\varepsilon = 0.5$ | $\varepsilon = 1$ | $\varepsilon = 1.5$ | $\varepsilon = 2$ | $\varepsilon = 2.5$ | $\varepsilon = 3$ |
| PeleeNet+ColorPriors | × | 93.1 | 20.0 | 12.5 | 11.0 | 9.9 | 9.5 | 9.2 |
| | ✓ | 93.1 | 21.5 | 14.2 | 11.8 | 9.8 | 10.2 | 10.1 |

**Table 3.** Comparison with or without model options. among them, prior success rate (PSR) is a model selection scheme based on prior success rate, and Information Entropy (IE) is a model selection scheme based on information entropy (datasets:imagenet).

| Part Model | Model Selection | | TOP1 Recognition Rate under Different FGSM Interference Amplitudes (%) | | | | | | |
|---|---|---|---|---|---|---|---|---|---|
| | PSR | IE | $\varepsilon = 0$ | $\varepsilon = 0.5$ | $\varepsilon = 1$ | $\varepsilon = 1.5$ | $\varepsilon = 2$ | $\varepsilon = 2.5$ | $\varepsilon = 3$ |
| PeleeNet+ | ✓ | × | × | 93.1 | 21.5 | 14.2 | 11.8 | 9.8 | 10.2 | 10.1 |
| ColorPriors | ✓ | ✓ | × | 93.1 | 21.6 | 14.2 | 11.8 | 10.2 | 9.8 | 9.8 |
| | ✓ | × | ✓ | 93.1 | 19.7 | 12.2 | 10.8 | 10.0 | 9.3 | 9.4 |

**Table 4.** Ablation experiments. under different disturbances, the optimal value is displayed in boldface and the sub-optimal value is shown underlined (datasets:imagenet).

| Part Model | Model Selection | | TOP1 Recognition Rate under Different FGSM Interference Amplitudes (%) | | | | | | |
|---|---|---|---|---|---|---|---|---|---|
| | PSR | IE | $\varepsilon = 0$ | $\varepsilon = 0.5$ | $\varepsilon = 1$ | $\varepsilon = 1.5$ | $\varepsilon = 2$ | $\varepsilon = 2.5$ | $\varepsilon = 3$ |
| PeleeNet | × | × | × | 93.0 | 18.4 | 12.3 | 10.1 | 9.7 | 8.8 | 8.2 |
| | × | × | × | 93.1 | 20.0 | 12.5 | 11.0 | 9.9 | 9.5 | 9.2 |
| PeleeNet+ | ✓ | × | × | 93.1 | 21.5 | 14.2 | 11.8 | 9.8 | 10.2 | 10.1 |
| ColorPriors | ✓ | ✓ | × | 93.1 | 21.6 | 14.2 | 11.8 | 10.2 | 9.8 | 9.8 |
| | ✓ | × | ✓ | 93.1 | 19.7 | 12.2 | 10.8 | 10.0 | 9.3 | 9.4 |

## 5.4. Comparative Study

In order to verify the generalization and robustness of the color prior model, we select three commonly used image classification data sets and six classic classification networks for grouping experiments.

We conduct comparative experiments through two aspects:

- Does the color prior method have a defensive effect on different classification networks.
- Is the color prior method applicable to different datasets.

The results of comparative experiments are shown in Tables 5–7. It can be seen from the tables that: (1) For the classification networks mentioned above, the adoption of the color prior model improves the recognition rate, even when the interference intensity is increasing. Especially for cifiar-10, when the interference factor $\varepsilon$ is 1, the recognition rate of SqueezeNet is improved by the maximum 4.2 percents after using the color prior model. It can be seen that the color prior model is universal to different classification networks and can be combined with the respective preprocessing modules or adversarial training methods. (2) The experimental results in the above three datasets show that, the defense capability and performance of the classification networks are improved by adopting the color prior model compared with the original framework, which proves that the color prior model enhances the defense ability under different data sets and has strong versatility.

In order to verify the defense ability of the proposed method under different types of attacks, we add three other attack methods, including DeepFool [35], PGD [30] and BIM [5], and do comparative experiments on Imagenet dataset. These attack methods are implemented in the Torchatbacks Library, which is a very good lightweight adversarial sample library. In the experiments, the steps parameter of PGD is set to 3, and the parameters of other attack methods remain the default settings. In Table 8, it can be seen that our method has certain defense abilities under FGSM, PGD, BIM, and DeepFool attack methods, which indicates that the method has good versatility.

**Table 5.** Comparative experiments on different classification networks (datasets:imagenet).

| | Color Priorss | TOP1 Recognition Rate under Different FGSM Interference Amplitudes (%) | | | | | | |
|---|---|---|---|---|---|---|---|---|
| | | $\varepsilon = 0$ | $\varepsilon = 0.5$ | $\varepsilon = 1$ | $\varepsilon = 1.5$ | $\varepsilon = 2$ | $\varepsilon = 2.5$ | $\varepsilon = 3$ |
| ResNet50 | × | 97.6 | 67.9 | 58.6 | 52.0 | 46.6 | 42.7 | 38.7 |
| | ✓ | 97.7 | 68.4 | 59.9 | 52.4 | 47.8 | 43.5 | 39.4 |
| VGG16 | × | 95.2 | 33.1 | 22.6 | 18.1 | 16.0 | 14.4 | 13.4 |
| | ✓ | 95.3 | 33.9 | 25.2 | 20.4 | 17.3 | 15.8 | 14.0 |
| MobileNetV2 | × | 94.3 | 39.8 | 32.3 | 26.5 | 21.2 | 18.7 | 16.4 |
| | ✓ | 94.6 | 41.2 | 34.1 | 27.1 | 22.7 | 20.1 | 18.0 |
| DenseNet161 | × | 97.7 | 78.3 | 72.0 | 68.2 | 65.0 | 61.8 | 58.8 |
| | ✓ | 98.0 | 78.7 | 73.9 | 68.6 | 65.2 | 61.9 | 59.4 |
| InceptionV3 | × | 95.4 | 71.1 | 66.7 | 63.5 | 61.0 | 58.7 | 56.0 |
| | ✓ | 95.4 | 71.7 | 67.3 | 64.8 | 61.4 | 59.7 | 56.6 |
| SqueezeNet | × | 92.5 | 31.4 | 22.4 | 18.4 | 14.9 | 13.0 | 11.5 |
| | ✓ | 92.6 | 33.1 | 24.3 | 18.9 | 16.1 | 14.9 | 12.2 |

**Table 6.** Comparative experiments on different classification networks (datasets:CIFAR-10).

| | Color Priorss | TOP1 Recognition Rate under Different FGSM Interference Amplitudes (%) | | | | | | |
|---|---|---|---|---|---|---|---|---|
| | | $\varepsilon = 0$ | $\varepsilon = 0.5$ | $\varepsilon = 1$ | $\varepsilon = 1.5$ | $\varepsilon = 2$ | $\varepsilon = 2.5$ | $\varepsilon = 3$ |
| ResNet50 | × | 93.3 | 32.0 | 25.0 | 21.5 | 18.6 | 16.7 | 15.6 |
| | ✓ | 93.3 | 33.2 | 26.3 | 22.3 | 19.3 | 17.4 | 15.9 |
| VGG16 | × | 94.2 | 41.8 | 38.0 | 34.6 | 31.6 | 29.2 | 26.8 |
| | ✓ | 94.4 | 42.9 | 39.0 | 35.4 | 32.3 | 29.9 | 27.1 |
| MobileNetV2 | × | 92.6 | 26.4 | 20.8 | 17.8 | 15.8 | 14.4 | 13.6 |
| | ✓ | 92.6 | 27.1 | 21.4 | 18.4 | 16.6 | 15.1 | 14.1 |
| DenseNet161 | × | 92.7 | 30.3 | 24.3 | 20.7 | 18.3 | 16.9 | 15.7 |
| | ✓ | 93.1 | 31.8 | 25.4 | 21.4 | 19.0 | 17.3 | 16.3 |
| InceptionV3 | × | 92.7 | 31.1 | 27.0 | 24.4 | 22.1 | 20.8 | 18.8 |
| | ✓ | 92.7 | 32.9 | 27.9 | 25.1 | 23.2 | 21.8 | 19.7 |
| SqueezeNet | × | 93.9 | 27.9 | 20.9 | 17.4 | 15.3 | 14.0 | 13.2 |
| | ✓ | 93.9 | 31.8 | **25.1** | 20.8 | 18.7 | 17.3 | 16.5 |

**Table 7.** Comparative experiments on different classification networks (datasets:fashion-mnist).

| | Color Priorss | TOP1 Recognition Rate under Different FGSM Interference Amplitudes (%) | | | | | | |
|---|---|---|---|---|---|---|---|---|
| | | $\varepsilon = 0$ | $\varepsilon = 0.5$ | $\varepsilon = 1$ | $\varepsilon = 1.5$ | $\varepsilon = 2$ | $\varepsilon = 2.5$ | $\varepsilon = 3$ |
| ResNet50 | × | 94.0 | 27.2 | 24.9 | 24.2 | 23.7 | 23.1 | 22.6 |
| | ✓ | 94.0 | 28.9 | 28.1 | 27.3 | 26.5 | 25.9 | 25.2 |
| VGG16 | × | 93.0 | 45.1 | 42.7 | 40.6 | 38.9 | 37.3 | 35.5 |
| | ✓ | 93.2 | 46.3 | 43.4 | 41.4 | 39.4 | 37.3 | 35.8 |
| MobileNetV2 | × | 94.6 | 29.0 | 21.9 | 18.6 | 15.6 | 14.3 | 14.2 |
| | ✓ | 95.0 | 30.1 | 22.0 | 20.5 | 17.1 | 16.2 | 14.6 |
| DenseNet161 | × | 93.8 | 38.7 | 36.5 | 33 | 30.1 | 25.1 | 21.3 |
| | ✓ | 94.0 | 39.9 | 38.0 | 35.3 | 32.1 | 27.5 | 24.7 |
| InceptionV3 | × | 93.8 | 40.4 | 38.8 | 37.0 | 35.3 | 32.7 | 29.6 |
| | ✓ | 93.9 | 42.4 | 40.6 | 39.4 | 37.7 | 35.3 | 32.3 |
| SqueezeNet | × | 92.4 | 27.3 | 25.0 | 23.4 | 21.9 | 20.3 | 19.2 |
| | ✓ | 92.6 | 29.8 | 27.5 | 26.0 | 25.2 | 24.3 | 23.1 |

**Table 8.** Comparative experiments on different attack methods (datasets:imagenet).

| | Color Priorss | Clean | FGSM | PGD | BIM | DeepFool |
|---|---|---|---|---|---|---|
| ResNet50 | × | 97.6 | 74.7 | 18.1 | 64.2 | 39.8 |
| | ✓ | 97.7 | 75.9 | 21.3 | 66.7 | 41.2 |

## 6. Conclusions

This paper proposes an image classification method based on color prior model. The proposed color prior model is cognitive-driven and has no training parameters, thus it has strong generalization and can effectively defend against adversarial samples. The proposed method directly combines color priors with convolution network features without changing the network structure and its parameters of the CNN, and can be used in combination with the respective preprocessing modules or adversarial training methods. Experiments show that color priors can effectively improve the defense ability of CNN with different structures under different levels of interference against samples. In future work, we will explore more effective model selection schemes, and study the fusion strategies between color prior models and neural networks at different stages.

**Author Contributions:** Conceptualization, P.G., C.Z. and S.L.; methodology, P.G., C.Z., X.L. and S.L.; software, P.G. and J.W.; formal analysis, P.G., C.Z. and S.L.; investigation, P.G., X.L. and J.W.; resources, C.Z., X.L. and S.L.; data curation, P.G.; writing—original draft preparation, P.G.; writing—review and editing, C.Z., X.L. and S.L.; supervision, C.Z. and S.L.; project administration, C.Z., X.L.and S.L.; funding acquisition, X.L. and S.L. All authors have read and agreed to the published version of the manuscript.

**Funding:** This work is partly supported by National Natural Science Foundation of China (Grant No. U19B2033), National Key R&D Program(Grant No. 2019YFF0301801), Frontier Science and Technology Innovation Project(Grant No. 2019QY2404), and the Innovation Academy for Light-Duty Gas Turbine, Chinese Academy of Sciences, under Grant CXYJJ19-ZD-02.

**Conflicts of Interest:** The authors declare no conflict of interest.

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
