# Peer review of "Robust Image Classification with Cognitive-Driven Color Priors"

_electronics, doi:10.3390/electronics9111837_

Round 1
Reviewer 1 Report
Please remove the formulas in Fig. 1 upper row boxes.
It is agreat article, my acknowledgements!
Author Response
Thank you for your review, keeping these formulas makes it easier to understand the calculation process of the prior model.
Reviewer 2 Report
This paper reports a color prior model to improve the image classification robustness based on human cognitive science. The methods is compared with current popular neural network methods and identified some advantages. Therefore, worth publication with a few moderate revisions:
- The research can be contribute to big earth data analytics. So review the general background of image classification, especially under the big earth data analytics context with expansion of literature on this direction.
- The conclusion is too light, major methods and results finding should be reported here.
- The details of implementation is not enough, suggest to introduce a software architecture with, if possible, source code open for others to duplicate.
- Although authors claimed improvements on anti-interference ability, the experimental results didn't show significant improvement.
- Author mentioned color priors are indexed from the memory, would be good to introduce the details of this.
- Could authors comment or compare with on LSTM methods.
- The language is good in general, would suggest have it proofed by a native speaker to smooth out the presentation.
Author Response
Thank you for your review
(1)The research can be contribute to big earth data analytics. So review the general background of image classification, especially under the big earth data analytics context with expansion of literature on this direction.
Thank you for your guidance.
(2)The conclusion is too light, major methods and results finding should be reported here.
We rewrote the conclusion and reported the major methods and results finding.
(3)The details of implementation is not enough, suggest to introduce a software architecture with, if possible, source code open for others to duplicate.
The implementation of our code mainly refers to tensorvision and torchattacks libray.
(4)Although authors claimed improvements on anti-interference ability, the experimental results didn't show significant improvement.
We mainly provide a new way of defending against adversarial attacks, and we will conduct further experiments to improve the defense effect in the future.
(5)Author mentioned color priors are indexed from the memory, would be good to introduce the details of this.
The detailed description of color priors is given in parts 1, 3, and 4. Our main idea is to design a color prior model by simulating human memory and reflection characteristics.
(6)Could authors comment or compare with on LSTM methods.
Our method is mainly used to improve the robustness of CNN-based image classification methods and the ability to defend against adversarial samples. We will study methods such as LSTM in the future.
(7)The language is good in general, would suggest have it proofed by a native speaker to smooth out the presentation.
We have improved the language expression to make the paper more fluent.
Reviewer 3 Report
The authors provide a very interesting solution to the adversarial sample problem. The work is tested very thoroughly and the results are promising.
Author Response
Thanks for your review.
Reviewer 4 Report
To defend against adversarial perturbations, this paper proposes a CNN-based image classifier with cognitive-driven color priors.
The paper is well organized and fully describes the methods for computing the color prior model and combining the model with CNN for image classification.
However, in the experimental results, the improvement in the recognition rate is not significant. This is the main reason why this paper cannot be accepted.
In the experiments, only the FGSM method was used to generate adversarial samples. So, wonder if the proposed method is able to deal with different types of adversarial perturbations (in three categories in Section 2.2).
Author Response
Thanks for your review. In Section 5.4, we added three other types of attack methods to supplement the comparative experiments, including DeepFool, PGD and BIM. Experimental results show that our method can deal with different types of adversarial perturbations.
Round 2
Reviewer 4 Report
The main point of this paper is to build and use color prior models to defend against adversarial perturbations.
But the advantage of using the color prior model is not clearly shown in the experimental results. In other words, the improvement in the recognition rate is not significant.
In the previous review, I pointed out this weakness as the main reason why this paper cannot be accepted.
But in the revision, the authors did not address this point.
Author Response
Reply to reviewer 4:
The main point of this paper is to build and use color prior models to defend against adversarial perturbations.
But the advantage of using the color prior model is not clearly shown in the experimental results. In other words, the improvement in the recognition rate is not significant.
In the previous review, I pointed out this weakness as the main reason why this paper cannot be accepted.
But in the revision, the authors did not address this point.
Reply:
Thanks for your review. The main work of this paper is to propose a new method to defense against adversarial attacks from a new perspective. Our focus is to verify the effectiveness of the existing method, and we will continue to improve the method to further improve the performance in the future.